# Pixel3DMM: Versatile Screen-Space Priors for Single-Image 3D Face Reconstruction

**Simon Giebenhain**[1]   **Tobias Kirschstein**[1]   **Martin Rünz**[2]
**Lourdes Agapito**[3]   **Matthias Nießner**[1]

[1]Technical University of Munich    [2]Synthesia    [3]University College London

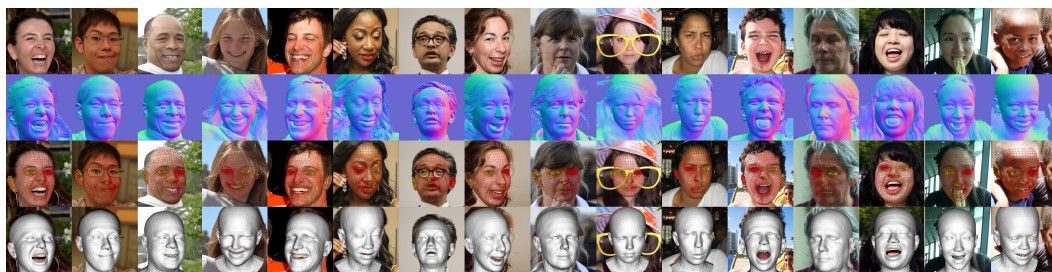

Figure 1: We present Pixel3DMM, a set of two ViTs Dosovitskiy et al. (2020), which are tailored to predict per-pixel surface normals and uv-coordinates. Here, we demonstrate the fidelity and robustness of Pixel3DMM on challenging inputs. From top to bottom we show input RGB, predicted normals, 2D vertices extracted from the uv-coordinate prediction, and our FLAME fitting results.

## Abstract

We address the 3D reconstruction of human faces from a single RGB image. To this end, we propose Pixel3DMM, a set of highly-generalized vision transformers which predict per-pixel geometric cues in order to constrain the optimization of a 3D morphable face model (3DMM). We exploit the latent features of the DINO foundation model, and introduce a tailored surface normal and uv-coordinate prediction head. We train our model by registering three high-quality 3D face datasets against the FLAME mesh topology, which results in a total of over 1,000 identities and 976K images. For 3D face reconstruction, we propose a FLAME fitting opitmization that solves for the 3DMM parameters from the uv-coordinate and normal estimates. To evaluate our method, we introduce a new benchmark for single-image face reconstruction, which features high diversity facial expressions, viewing angles, and ethnicities. Crucially, our benchmark is the first to evaluate both posed and neutral facial geometry. Ultimately, our method outperforms the state-of-the-art (SoTA) by over 15% in terms of geometric accuracy for posed facial expressions.

## 1 Introduction

3D reconstruction of faces, tracking facial movements, and ultimately extracting expressions for animation tasks are fundamental problems in many domains such as computer games, movie production, telecommunication, and AR/VR applications. Recovering 3D head geometry from a single image is a particularly important task due to the vast amount of available image collections.

Unfortunately, reconstructing faces from a single input image is also inherently under-constrained. Not only depth ambiguity renders this task challenging, but also ambiguities between albedo and lighting/shadow effects. In addition, properly disentangling identity and expression information – which is critical for many downstream applications – makes the problem difficult. Finally, occlusions and unobserved facial regions further complicate the problem in real application scenarios, thus highlighting the need for strong data priors.

A typical approach to single-image face reconstruction is to exploit 3D parametric head models (3DMMs) Blanz & Vetter (2023); Li et al. (2017) which provide a low-dimensional

parametric representation for the underlying 3D geometry. Optimizing within a 3DMM's disentangled parameter space heavily constrains the search space with built-in assumptions about plausible facial structure and expressions, and allows to extract disentangled identity and expression information. Nonetheless, despite relying on 3DMMs, many ambiguities remain and their simplifying assumptions about our world often cannot explain the complexity of the real world. This necessitates additional priors in order to obtain compelling fitting results such as sparse Sagonas et al. (2013) and dense Cao et al. (2013); Wood et al. (2022) facial landmarks, or UV coordinate predictions Taubner et al. (2024a).

In recent years, we have also seen significant progress in feed-forward 3DMM regressors Sanyal et al. (2019); Feng et al. (2021); Daněček et al. (2022); Retsinas et al. (2024); Zielonka et al. (2022); Zhang et al. (2023). However, it is complicated to extend feed-forward regressors, *e.g.* to a multi-view or temporal domain, and, as we will show later, they fall behind optimization-based approaches on inputs with strong facial expressions. Overall, accurate 3D face reconstruction from single images remains a challenging and highly relevant problem.

Therefore, we propose Pixel3DMM, a novel optimization-based 3D face reconstruction approach. Our main idea is to exploit and further develop broadly generalized and powerful foundation models to predict pixel-aligned geometric cues that effectively constrain the 3D state of an observed face. Given a single image at test time, we propose normal and uv-coordinate predictions as optimization constraints from which we fit a 3D FLAME model. Instead of a simple rendering loss of uv-coordinates, we then transfer the information into a 2D vertex loss, which offers a wider basin of attraction during optimization. We argue that this strategy is superior to traditional photometric terms, or sparse landmarks, which often struggle with extreme view points and facial expressions. In order to train our approach, we unify three recent, high-fidelity 3D face datasets Giebenhain et al. (2023); Zhu et al. (2023); Martinez et al. (2024) by registering them against the FLAME Li et al. (2017) model.Our approach outperforms all available normal estimators for human faces in the NeRSemble Kirschstein et al. (2023) dataset.

In order to advance the evaluation of single-image 3D face reconstruction methods, we further propose a new benchmark based on the multi-view video dataset NeRSemble Kirschstein et al. (2023), which includes a wider variety of facial expressions than existing benchmarks Sanyal et al. (2019); Zhu et al. (2023); Feng et al. (2018); Chai et al. (2022). Our benchmark is the first to allow for the simultaneous evaluation of posed and neutral facial geometry. This enables a more direct comparison of methods, especially regarding fitting fidelity and ability to disentangle expression and identity information. Finally, we show that compared to our strongest baselines, our approach improves the L2-Chamfer reconstructions loss by over 15% for posed geometry, while slightly improving over neutral geometry predictions.

To summarize, our main contributions are as follows:
- A new formulation to exploit foundation model features for 3D-related, pixel-aligned predictions, facilitating SoTA normal estimations for human faces.
- A novel 3D face reconstruction approach based on predicted uv-map correspondences and surface normals.
- A 3D face reconstruction benchmark and evaluation protocol from high-fidelity multi-view face captures.

We plan to make the model, code, and our new benchmark publicly available to promote progress in single image 3D face reconstruction and encourage quantitative benchmarking on challenging facial expressions.

## 2 RELATED WORK

**Single-Image 3DMM Fitting**  Tracking morphable models from single images is a well-studied problem in the context of 3D face reconstruction and tracking. Early works Blanz & Vetter (1999); Paysan et al. (2009); Li et al. (2017), introduced statistical shape and texture priors to estimate 3D face geometry from 2D images. Such methods rely on photometric fitting and subsequent approaches improve modeling capabilities using learned implicit

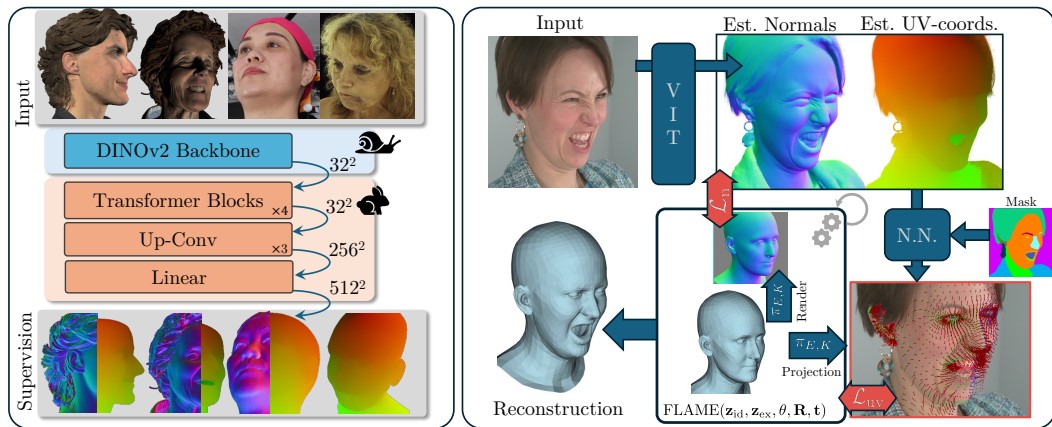

Figure 2: **Method Overview:** Pixel3DMM consists of (a) learning pixel-aligned geometric priors (left) and (b) test-time optimization against predicted uv-coordinates and normals (right). On the left we show our network architecture and training examples. On the right we depict the process of finding per-vertex 2D locations using a nearest neighbor (N.N.) look up, and our loss terms.

representations Lin et al. (2023); Giebenhain et al. (2024). While some methods Thies et al. (2016); Grishchenko et al. (2020) favor a high tracking frame rate for real-time applications, others favor reconstruction accuracy Zielonka et al. (2022).

**Facial Landmark Prediction** Numerous reconstruction methods Li et al. (2017); Cao et al. (2013) for faces rely on accurate landmark predictions, which are usually coupled with vertices of a template mesh. Pioneering work on detecting such landmarks already relies on statistical learning Cootes et al. (2001) and more recent models exploit large datasets Wood et al. (2021); Wu et al. (2018) and neural networks to improve the performance Bulat & Tzimiropoulos (2017); Bazarevsky et al. (2019). MediaPipe Bazarevsky et al. (2019), for instance, uses a convolutional network inspired by MobileNet Howard et al. (2017). Another line of work focuses on densely aligning template mesh and 2D predictions. To achieve this FlowFace Taubner et al. (2024a) employs a vision-transformer backbone and iteratively refines the flow from UV to image space.

**3DMM Regression** DECA Feng et al. (2021) trains a regressor to predict 3DMM parameters from an image . An extension of this work is presented in EMOCA Daněček et al. (2022), which emphasizes the reconstruction of emotion-rich expressions. Similarly, SPECTRE Filntisis et al. (2022) aims at temporal consistency and reconstructing lip motion. SMIRK Retsinas et al. (2024) introduces a neural synthesis component, reducing the domain gap between real and rendered images. Since the aforementioned methods don't assume 3D training data, it is easy to scale them to large datasets. As a downside, the lack of 3D information impedes accuracy and leaves depth ambiguity. In order to address this, MICA Zielonka et al. (2022) supervises directly in 3D space. TokenFace Zhang et al. (2023) is a transformer-based method that can be trained on both 2D and 3D data.

**Face Reconstruction Benchmarks** The Stirling Feng et al. (2018) dataset contains 2000 images of 135 subjects. Unfortunately, ground truth reconstructions are only available for neutral poses in this dataset. Similarly, the NoW Sanyal et al. (2019) benchmark provides reconstructions only in the neutral expression. It has 2054 images of 100 subjects.Both the FaceScape Zhu et al. (2023) and the REALY Chai et al. (2022) dataset contain posed scans. While the former has 10 identities, the latter has 100 subjects. Neither of these two benchmarks measures disentanglement by additionally evaluating against neutral geometry.

## 3 PIXEL3DMM

In this work we address the challenges of single-image face reconstruction by learning powerful priors of pixel-aligned geometric cues. In particular we train two vision transformer networks, which predict uv-coordinates and surface normals against which we

fit FLAME Li et al. (2017) parameters at inference time. In section 3.1 we describe our Pixel3DMM networks, our data acquisition, and how we train them for accurate surface normal and uv-coordinate prediction. Afterwards, in section 3.2, we elaborate on our single-image fitting approach, which is purely based on our surface normal and uv-coordinate predictions.

## 3.1 Learning Pixel-Aligned Geometric Cues

Despite recently released high-quality 3D face datasets Zhu et al. (2023); Kirschstein et al. (2023); Giebenhain et al. (2023); Martinez et al. (2024), such data is still relatively scarce, especially w.r.t. the number of different identities, ethnicities, age distribution and lighting variation. We therefore take inspiration from recent achievements on fine-tuning foundational and large generative models to become experts on a constrained domain, *e.g.* Hu et al. (2022); Ruiz et al. (2023).

In particular we train two expert networks

$$\mathcal{N} : \mathbb{R}^{512 \times 512 \times 3} \to [-1, 1]^{512 \times 512 \times 3} \quad \text{and} \quad \mathcal{U} : \mathbb{R}^{512 \times 512 \times 3} \to [\, 0, 1]^{512 \times 512 \times 2} \qquad (1)$$

which, given a single input image $I$, predict surface normals $\mathcal{N}(I)$ and uv-space coordinates $\mathcal{U}(I)$, respectively.

### 3.1.1 Network Architecture

We build Pixel3DMM on top of the foundational features from a pre-trained DINOv2 Oquab et al. (2023) backbone. As depicted in fig. 2, we extend the ViT architecture using a simple prediction head. It consists of four additional transformer blocks, three up-convolutions which lift the feature map resolution from 32 to $256 \times 256$. Finally, we use a single linear layer to increase the feature dimensionality and unpatchify the predictions to $512 \times 512 \times c$, where $c \in \{3, 2\}$ for normals and uv-coordinate prediction tasks, respectively.

### 3.1.2 Data Preparation

To train our networks, we opt for three recent, high-quality 3D face datasets: NPHM Giebenhain et al. (2023), FaceScape Zhu et al. (2023), and Ava256 Martinez et al. (2024). We follow the non-rigid registration procedure from NPHM, register all datasets into a uniform format and topology. fig. 2 shows pairs of input views with the associated supervision signal for surface normals and uv-coordinates.

**Dataset Numbers** In total, our dataset comprises 470 identities from NPHM in 23 expression and 40 renderings each (376K rgb, normal and uv images in total). For FaceScape we use 350 subjects, observed under 20 different expressions and 50 cameras each (350K rgb, normal and uv images in total). Since Ava256 is a video dataset, we leverage furthest point sampling to select the 50 most diverse expressions per person. For each person we choose a random subset of 20 cameras (250K rgb and uv images in total).

**Diffsion-based Lighting Variations** Since FaceScape and Ava256 are both studio datasets, which are captured at rather homogeneous lighting conditions, we leverage IC-Light Zhang et al. (2025), an image conditioned diffusion model Rombach et al. (2022), which alters the lighting condition based on a text prompt or background image.

### 3.1.3 Training

We train our models $\mathcal{M} \in \{\mathcal{N}, \mathcal{U}\}$ using a straight forward image translation formulation

$$\underset{\Psi_{\mathcal{M}}}{\arg\min} \sum_{k \in \mathcal{D}} \sum_{p \in M^k} \|f(I^k)_p - Y_p^k\|_2, \qquad (2)$$

where $\Psi_{\mathcal{M}}$ denotes the network's parameters, $k \in \mathcal{D}$ is a sample from our dataset, $I^k$ and $Y^k$ are input rgb and target images, respectively, and $p \in M^k$ are all pixels in the associated foreground mask.

Note, that instead of freezing the parameters of our DINOv2 backbone altogether, we set their learning rate ten times lower, in order to encourage prior preservation but enable stronger domain adoption.

Compared to Sapiens Khirodkar et al. (2024), a recent SoTA foundation model for human bodies and faces, training our models is cheap and can be realized using 2 GPUs and training for 3 days. Additionally, all our data is publicly available. The relatively low computational burden and data accessibility, will hopefully inspire more research to follow in a similar direction.

## 3.2 SINGLE-IMAGE FLAME FITTING

Given a single image $I$, we leverage our prior networks to obtain predicted surface normals $\mathcal{N}(I)$ and uv-coordinates $\mathcal{U}(I)$. Using these predictions we aim to recover 3DMM parameters. In particular, we optimize for FLAME Li et al. (2017) identity, expression, and jaw parameters, as well as, camera rotation, translation, focal length and principal point:

$$\Omega_{\text{FLAME}} = \{\mathbf{z}_{\text{id}} \in \mathbb{R}^{300}, \mathbf{z}_{\text{ex}} \in \mathbb{R}^{100}, \theta \in \mathcal{SO}(3)\} \tag{3}$$

$$\Omega_{\text{cam}} = \{\mathbf{R} \in \mathcal{SO}(3), \mathbf{t} \in \mathbb{R}^3, \mathbf{fl} \in \mathbb{R}^+, \mathbf{pp} \in \mathbb{R}^2\}. \tag{4}$$

### 3.2.1 2D VERTEX LOSS

Using the estimated uv-coordinates $\mathcal{U}(I)$, we aim to extract the 2d location $p_v^*$ for each visible vertex $v \in V$ of the FLAME mesh. To this end we first run a facial segmentation network Zheng et al. (2022), in order to mask out the background, eyeballs and mouth interior. Then we find correspondences for each vertex $v \in V$ using a nearest neighbor lookup into $\mathcal{U}(I)$. To be more specific let $T_v^{\text{uv}} \in [0,1]^2$ denote the uv-coordinate of $v$ in the template mesh $T$. Then we find the pixel location

$$p_v^* = \underset{p \in P}{\arg\min} \|T_v^{\text{uv}} - \mathcal{U}(I)_p\| \tag{5}$$

as the pixel with the closest uv prediction. Finally, we define

$$\mathcal{L}_{\text{uv}} = \sum_{v \in V} \mathbb{1}_{\|T_v^{\text{uv}} - \mathcal{U}(I)_p\| < \delta_{\text{uv}}} \cdot |p_v^* - \pi(v)| \tag{6}$$

to be our 2d vertex loss, where $\mathbb{1}$ denotes the indicator function which masks out vertices with a nearest neighbor distance larger than $\delta_{uv}$. $V = \text{FLAME}(\Omega_{\text{FLAME}})$ is the current estimate of the FLAME parametric model, and $\pi$ denotes the projection implied by the current estimate of the camera parameters $\Omega_{\text{cam}}$.

### 3.2.2 OPTIMIZATION

Next to the 2d vertex loss $\mathcal{L}_{uv}$, we include the normal loss $\mathcal{L}_n = |\mathcal{N}(I) - \text{render}_n(V)|$, where $\text{render}_n$ denotes a rendering of surface normals of the FLAME mesh. The regularization term $\mathcal{R} = \lambda_{\text{id}} \|\mathbf{z}_{\text{id}} - \mathbf{z}_{\text{id}}^{\text{MICA}}\|_2^2 + \lambda_{\text{ex}} \|\mathbf{z}_{\text{ex}}\|_2^2$ completes our overall energy term

$$E = \lambda_{uv} \mathcal{L}_{uv} + \lambda_n \mathcal{L}_n + \mathcal{R}. \tag{7}$$

Here $\mathbf{z}_{\text{id}}^{\text{MICA}}$ denotes MICA's Zielonka et al. (2022) identity prediction.

## 3.3 MONOCULAR VIDEO TRACKING

Next to the single-image scenario, tracking faces in monocular videos is a fundamental task in computer vision. To address this problem, we simply extend our optimization strategy from section 3.2.2 globally over all images in a video sequence $\{I_t\}_{t=1}^T$. Using our prior networks, we first obtain normal predictions $\{\mathcal{N}(I_t)\}$ and uv-predictions $\{\mathcal{U}(I_t)\}$ After obtaining an initial estimate for $\Omega_{\text{FLAME}}^{(0)}$ and $\Omega_{\text{cam}}^{(0)}$ on the first frame by optimizing for eq. (7), we freeze $\mathbf{z}_{\text{id}}$, $\mathbf{fl}$ and $\mathbf{pp}$. We then sequentially optimize for all remaining attributes in $\Omega_{\text{FLAME}}^{(t)}$ and $\Omega_{\text{cam}}^{(t)}$. Using the results from the sequential optimization pass as initialization,

we extend eq. (7) to a batched version using randomly sampled frames. Note, that the parameters $\mathbf{z}_{\text{id}}$, $\mathbf{fl}$ and $\mathbf{pp}$ are shared for all frames. In order to enforce smoothness across all per-frame optimization targets we add a smoothness term

$$\mathcal{L}_{\text{smooth}}^{\Phi} = \frac{\lambda_{\text{smooth}}^{\Phi}}{2 * B} \sum_{t \in B} \|\Phi^{(t-1)} - \Phi^{(t)}\|_2^2 + \|\Phi^{(t)} - \Phi^{(t+1)}\|_2^2 \tag{8}$$

to the energy $E$, where $\Phi^{(t)} \in \{\mathbf{z}_{\text{ex}}^{(t)}, \theta^{(t)}, \mathbf{R}^{(t)}, \mathbf{t}^{(t)}\}$ denotes any of the per-frame variables.

## 4 3D FACE RECONSTRUCTION BENCHMARK

Constructing a benchmark that covers the variety of facial geometry and its complex deformations is a challenging endeavor. In table 1 we present a comparison of popular single-image face reconstruction benchmarks.

Table 1: **Comparison of 3D Face Reconstruction Benchmarks.** We compare data capture year, whether the benchmark evaluates posed and/or neutral geometry, expression diversity, viewpoint diversity, number of persons (#pers.) and number of GT scans.

| | Year | posed | neutral | expression diversity | viewpoint diversity | #pers. | #Scans |
|---|---|---|---|---|---|---|---|
| Stirling Feng et al. (2018) | 2013 | | ✓ | | ✓ | **133** | 133 |
| REALY Chai et al. (2022) | 2015 | ✓ | | | | 100 | 100 |
| NoW Sanyal et al. (2019) | 2019 | | ✓ | | ✓ | 80 | 80 |
| FaceScape Zhu et al. (2023) | 2020 | ✓ | | ✓ | ✓ | 20 | 20 |
| Ours | 2023 | ✓ | ✓ | ✓ | ✓ | 21 | **441** |

Compared to existing benchmark, ours focuses on strong facial expressions, as shown in fig. 3, where we retrieve the 5 most expressive images from the recent FaceScape benchmark Zhu et al. (2023) and the established NoW benchmark Sanyal et al. (2019). We do this by running EMOCA Daněček et al. (2022) on each image of the dataset, collecting the expression codes, and then performing furthest point sampling in EMOCA's expression space, starting from the expression with highest norm.

Furthermore, our benchmark is the first to jointly evaluate *posed* and *neutral* face reconstruction. Our benchmark contains 21 subjects, each in its neutral state and in 20 different and diverse facial expressions. We e hope that our proposed benchmark will be adopted as a standard by the community to encourage better quantitative comparisons across methods. For more information we refer to appendix A.1.

### 4.1 TASK DESCRIPTION AND EVALUATION PROTOCOL

Our benchmark consists of *posed* and *neutral* 3D face reconstruction. The posed reconstruction task aims to measure the fidelity of a 3D reconstruction. Given any expressive face image, the underlying geometry shall be recovered. The neutral reconstruction task measures how well a reconstruction method can disentangle the effects of shape and expression. Specifically, the task is to reconstruct the face under neutral expression given an image of the person under any arbitrary expression. Both tasks are evaluated using standard practice, and refer to appendix A.1 for more details.

## 5 EXPERIMENTAL RESULTS

### 5.1 IMPLEMENTATION DETAILS

**Prior Learning** We train Pixel3DMM using the Adam Kingma & Ba (2014) optimizer, a batch size of $40$, and 2 A6000 GPUs, which takes 3 days until convergence. We use a learning rate of $1 \times 10^{-4}$ for the prediction head and $1 \times 10^{-5}$ for the DINO backbone. For simplicity we choose a light-weight network head. Using a DPT Ranftl et al. (2021) head instead

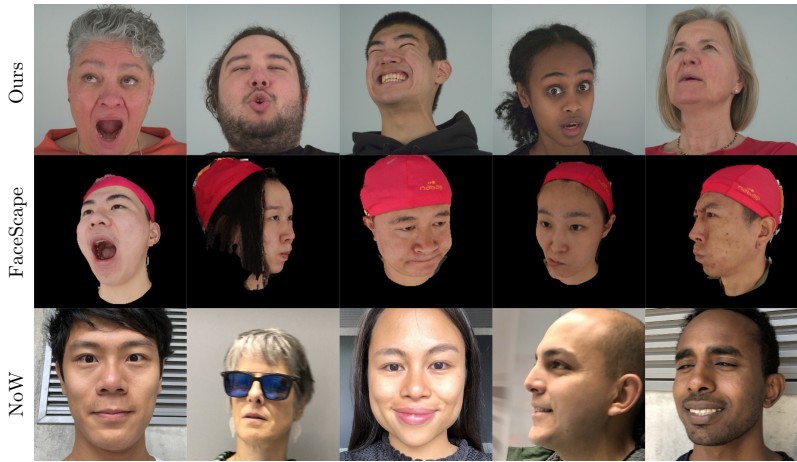

Figure 3: **3D Face Reconstruction Benchmark Analysis.** We show the 5 most diverse images from each benchmark dataset, as measured by the expression codes of EMOCA Daněček et al. (2022). Our benchmark covers a richer diversity of facial expressions.

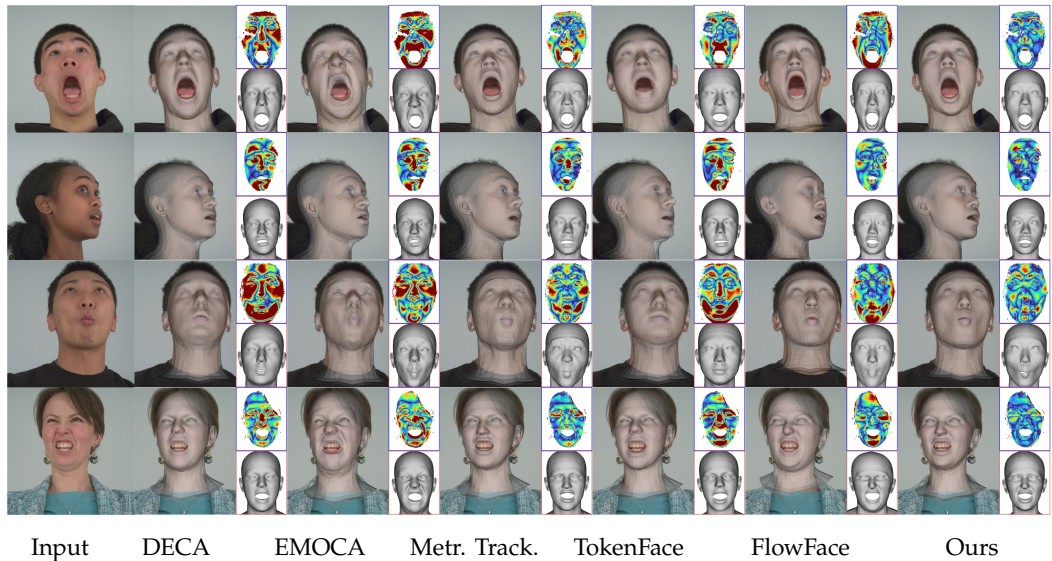

| Input | DECA | EMOCA | Metr. Track. | TokenFace | FlowFace | Ours |

Figure 4: **Qualitative Comparison (Posed):** We show overlays of the reconstructed meshes. Insets with a blue border depict $L_2$-Chamfer distance as an error map, rendered from a frontal camera. Red insets show the reconstructed mesh from the same camera. We encourage the reviewers to watch our supplementary material for additional visualizations.

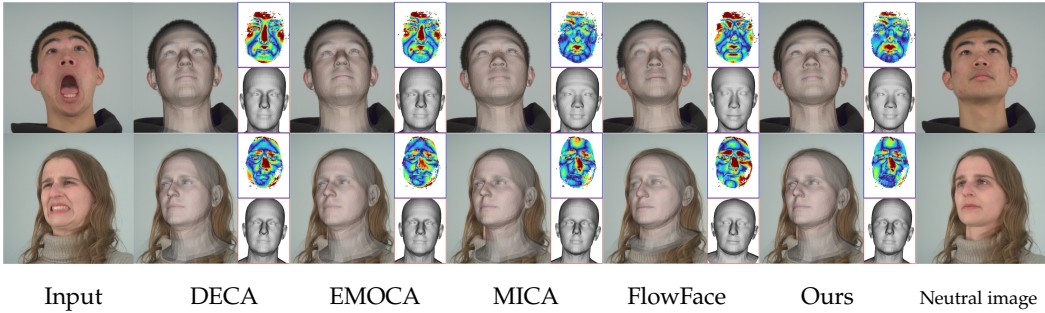

| Input | DECA | EMOCA | MICA | FlowFace | Ours | Neutral image |

Figure 5: **Qualitative Comparison (Neutral):** Alignment against the neutral expression.

resolves the last remaining patch artifacts of the ViT-Base backbone but drastically increases

Table 2: **Results on our benchmark.**

|  | Neutral | | | Posed | | |
|---|---|---|---|---|---|---|
|  | L1↓ | NC↑ | $R^{2.5}$↑ | L1↓ | NC↑ | $R^{2.5}$↑ |
| MICA | 1.68 | **88.3** | 91.0 | - | - | - |
| TokenFace | - | - | - | 2.62 | 86.5 | 76.8 |
| DECA | 2.07 | 87.6 | 84.5 | 2.38 | 87.0 | 79.8 |
| EMOCAv2 | 2.21 | 87.3 | 82.4 | 2.63 | 86.0 | 75.8 |
| Metr. Tracker | - | - | - | 2.03 | 87.8 | 85.7 |
| NHA (stage1) | 2.35 | 86.9 | 80.6 | 2.67 | 86.4 | 76.2 |
| VHAP (stage1) | 2.95 | 84.7 | 71.0 | 3.04 | 84.8 | 69.9 |
| FlowFace | 1.93 | 87.8 | 87.0 | 1.96 | 87.9 | 87.9 |
| Ours | **1.66** | **88.3** | **91.2** | **1.66** | **88.4** | **91.6** |

Table 3: **Existing benchmarks.**

| Method | NoW | | FaceScape | | |
|---|---|---|---|---|---|
|  | Med.↓ | Mean↓ | CD↓ | MNE↓ | CR↑ |
| Dense | 1.02 | 1.28 | - | - | - |
| PRNet | - | - | 3.56 | .126 | 89.6 |
| 3DDFAv2 | - | - | 3.60 | .096 | 93.1 |
| DECA | 1.09 | 1.38 | 4.69 | .108 | **99.5** |
| MICA | 0.90 | 1.11 | - | - | - |
| FlowFace | 0.87 | 1.07 | 2.21 | .083 | - |
| TokenFace | **0.76** | **0.82** | 3.70 | .101 | 93.8 |
| Ours | 0.87 | 1.07 | **1.76** | **.077** | 98.0 |

runtime whithout improving down-stream reconstruction performance. Similarly, we find that replacing ViT-Base with Sapiens-300M Khirodkar et al. (2024) backbone (the smallest available Sapiens model) incurs high computational costs without reconstruction benefits. We use 10% of the subjects as validation set, and exclude all the subjects from our benchmark from the training set.

**FLAME Fitting** We use the Adam optimizer with $lr_{id} = 0.001$ and $lr_{ex} = 0.003$. We set $\lambda_{uv} = 2000$, $\lambda_n = 200$, $\lambda_{id} = 0.15$ and $\lambda_{ex} = 0.01$. We perform 500 optimization steps which takes 30 seconds in our unoptimized implementation. As a comparison, the widely established MetricalTracher Zielonka et al. (2022) operates at roughly 2 frames per minute for their online-tracking approach, while our method achieves a total runtime of 30 frames per minute (measure and averaged over a video with 300 frames). All runtime measurements were performed on an RTX3080 GPU.

## 5.2 BASELINES

**Feed-Forward FLAME Regressors** The first category of approaches we compare against are feed-forward neural networks trained to predict FLAME parameters. In this category of baselines, we choose DECA Feng et al. (2021) and EMOCA Daněček et al. (2022) which are trained on 2D data only. Additionally, we compare against MICA Zielonka et al. (2022), which is trained on 3D data and only predicts identity parameters $z_{id}$, and TokenFace Zhang et al. (2023) which istrained on a mixture of 2D and 3D data.

**Optimization-Based Approaches** We compare against MetricalTracker Zielonka et al. (2022), which optimizes against two sets of facial landmark predictions Bulat & Tzimiropoulos (2017); Cao et al. (2013) and a photometric term. Additionally, we compare against Flow-Face Taubner et al. (2024a), a recent method that predicts flow from the uv-space into image space, in order to predict 2D image-space vertex positions. Similar to Pixel3DMM, FlowFace also uses a dense 2D vertex loss, but predicts them in a quite different manner. Finally, we compare against VHAP (Qian et al., 2024) and Neural Head Avatars (NHA) (Grassal et al., 2022), which start by optimizing within FLAME space (stage1) and continue by optimizing for vertex offsets (stage2).

**Normal Estimation** We compare against the industry-born normal estimator Sapiens-2B (Khirodkar et al., 2024) and concurrent work DAViD (Saleh et al., 2025). We also compare against Deep Face Normals (DFN) (Abrevaya et al., 2020) and Diff-E2E (Martin Garcia et al., 2025), which distill an estimator from StableDiffusion (Rombach et al., 2022)

## 5.3 OUR BENCHMARK

**Posed Face Reconstruction** We present quantitative and qualitative results for the posed reconstruction task (see section 4.1) in table 2 and fig. 4, respectively. Quantitatively, Pixel3DMM outperforms all baselines by a large margin. In general, the feed-forward predictors (DECA, EMOCAv2, TokenFace) perform significantly worse than the optimization based approaches (MetricalTracker, FlowFace and Ours). Visually, DECA and TokenFace seem to underfit facial expressions, while EMOCAv2 exaggerates them. Compared to our approach, FlowFace sometimes exhibits performance drops for extreme facial expressions.

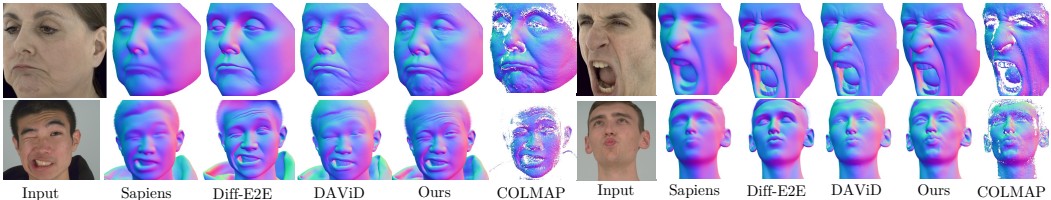

Input    Sapiens    Diff-E2E    DAViD    Ours    COLMAP    Input    Sapiens    Diff-E2E    DAViD    Ours    COLMAP

Figure 6: **Surface Normal Estimation:** Qualitative comparison to SoTA surface normal estimators.

Table 4: **Ablation Study.**

|  | Neutral | | | Posed | | |
|---|---|---|---|---|---|---|
|  | L1↓ | L2↓ | R$^{2.5}$↑ | L1↓ | L2↓ | R$^{2.5}$↑ |
| Lmks. | 1.68 | 1.14 | 91.1 | 2.02 | 1.37 | 85.7 |
| Lmks.+Pho. | 1.69 | 1.14 | 90.8 | 2.05 | 1.38 | 85.4 |
| Ours+Lmks.+Pho. | 1.68 | 1.14 | 91.0 | 1.86 | 1.26 | 88.3 |
| only $\mathcal{U}$ | **1.66** | **1.11** | **91.3** | 1.72 | 1.16 | 90.6 |
| only $\mathcal{N}$ | 1.69 | 1.12 | 90.7 | 1.70 | 1.14 | 91.0 |
| only Sapiens | 1.72 | 1.16 | 90.2 | 1.81 | 1.23 | 89.0 |
| Ours | **1.66** | 1.12 | 91.2 | **1.66** | **1.11** | **91.6** |
| no MICA | 1.90 | 1.29 | 87.2 | 1.74 | 1.17 | 90.1 |

Table 5: **Normal Estimation.**

| Method | H3DS | MultiFace | NeRSemble |
|---|---|---|---|
| DFN | 0.878 | 0.914 | 0.907 |
| Diff-E2E | 0.889 | 0.933 | 0.911 |
| sapiens | 0.902 | 0.950 | 0.911 |
| David | 0.903 | 0.943 | 0.927 |
| Ours | **0.905** | **0.958** | **0.931** |
| Ours* | 0.912 | 0.962 | 0.934 |

**Neutral Face Reconstruction**    Results on the neutral reconstruction task (see section 4.1) are provided in fig. 5 and table 2. First of all, we can observe that the significantly better posed reconstruction metrics of FlowFace and Pixel3DMM do not immediately translate to the neutral reconstruction. We attribute this to the ambiguities between identity and expression in the optimization process. Note that both FlowFace and Pixel3DMM rely on MICA predictions to initialize identity parameters $z_{id}$. While FlowFace ends up with worse neutral reconstructions, our approach is able to improve upon MICA by a small margin. Nevertheless, we highlight the importance of using MICA to help disambiguate between $z_{id}$ and $z_{ex}$, as ablated in section 5.7. Note, that TokenFace is missing from the neutral evaluation, since TokenFace's authors only provided posed meshes.

### 5.4    Results on Existing Benchmarks

**FaceScape Benchmark Zhu et al. (2023)**    The FaceScape benchmark only evaluates the posed reconstruction task. The relative performance across methods matches with results on our benchmark, see table 3. Our method outperforms all baselines by a large margin w.r.t. chamfer distance (CD) and mean normal error (MNE), and has a slightly worse completeness rate (CR) than DECA, see Zhu et al. (2023) for more details.

**NoW Benchmark Sanyal et al. (2019)**    On the NoW benchmark, which only evaluates the neutral reconstruction task, we achieve the same metrics as FlowFace, which is the best-performing optimization-based approach, but perform worse than TokenFace. However on FaceScape and our benchmark we significantly outperform TokenFace. Similarly to the results on our benchmark, Pixel3DMM can only improve a small amount on top of the MICA predictions. We hypothesize that our prior significantly helps posed reconstructions, but struggles to guide the optimization to properly disentangle between $z_{id}$ and $z_{ex}$.

### 5.5    In-the-Wild Results

In fig. 1, we demonstrate the robustness of our prior networks and fitting algorithm on challenging in-the-wild examples, including strong appearance variation, various background contexts and surroundings, lighting/shadow effects, and occlusions such as glasses, head wear and hands. Ultimately, this demonstrates that our approach successfully generalizes, beyond the training data distribution. We hope that this will inspire more work in a similar direction, especially since all data is available and 2 48GB GPUs are sufficient for training. For tracking results on in-the-wild monocular videos we refer to our supplementary video.

## 5.6 Surface Normal Estimation

In table 5 and fig. 6, we show quantitative and qualitative comparisons against recent state-of-the-art normal estimation methods. Our network estimates more detailed and accurate normals than the baselines. DAViD, a concurrent work to ours train on a vast synthetic dataset, predicts the most competitive results. However, DAViD struggles to accurately predict skin creases caused by complex deformations (see fig. 6), highlighting the need for real data. We also train a version of Pixel3DMM on the union of our data and the DAViD data, denoted as "Ours*". See appendix A.4 for more qualitative in-the-wild results.

## 5.7 Ablation Experiments

We conduct extensive ablations on different compositions of our optimization energy $E$ in table 4. We start by using the simplest energy, with only the landmark loss from MetricalTracker, and our regularization term. Next we add a photometric term, as in MetricalTracker. As shown in table 4, these configurations achieve significantly worse posed reconstructions. Interestingly, adding landmarks and photometric terms to the complete our proposed energy deteriorates reconstruction performance. Next, we investigate the effect of only using the predictions from $\mathcal{N}$ and $\mathcal{U}$, respectively. Compared to our full model these variants showcase lower posed reconstruction scores. We also compare our normal predictor $\mathcal{N}$ against Sapiens-2B Khirodkar et al. (2024), which confirms that our improved normal predictions translate to better reconstructions. Finally, we ablate the effect of using MICA. Without MICA's predictions of $\mathbf{z}_{id}$ especially the neutral reconstruction metrics drop, indicating its importance for disentanglement between identity and expression.

## 5.8 Additional Results

We highly encourage the reviewers to watch our supplementary video, and qualitative video tracking comparisons against the most competitive baseline (as suggested by our benchmark), which has publicly available code.

## 6 Limitations and Future Work

While we demonstrate the effectiveness of our approach for single image 3D reconstruction, several limitations remain. While our optimization energy could be easily extended to incorporate observations from multiple viewpoints, our prior models cannot currently exploit multiview information. Future extensions of our architecture could include multiview inputs similar to DUSt3R Wang et al. (2024), or video inputs similar to RollingDepth Ke et al. (2024). Next, for training large-scale 3DMM conditioned generative models like 3D GANS Sun et al. (2023) or diffusion models Kirschstein et al. (2024); Prinzler et al. (2024); Taubner et al. (2024b), e.g. on the LAION-Face dataset Zheng et al. (2022), fast reconstruction speed would be desirable. One potential avenue could be the distillation of our per-pixel predictors into a feed-forward 3DMM predictor. Finally, our experiments showcase, that optimization based approaches cannot flawlessly disambiguate identity and expression parameters. Therefore, specifically crafted priors for disambiguation are required.

## 7 Conclusion

In this paper, we trained pixel-aligned geometric prior networks, by leveraging pre-trained, generalized foundational features on publicly available 3D face datasets, which we registered into a uniform format. Our trained networks successfully generalize beyond the diversity of the training data, and we experimentally show that our normal predictor significantly outperforms all available normal estimators. We designed a 3DMM fitting algorithm on top of our prior predictions, which results in state of the art single image 3D reconstruction. Finally, we introduce a new benchmark, which features diverse and extreme expressions and allows, for the first time, to simultaneously evaluate neutral and posed geometry.

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

# A  Appendix

In this appendix we provide additional information about our benchmark (see appendix A.1), and additional qualitative results for normal estimation and two more baselines, in sections A.4 and A.5, respectively. Additionally, we highly encourage the reviewers to watch out supplemental video, including qualitative comparisons for video tracking.

## A.1  3D Face Reconstruction Benchmark

Human face geometry is complex due to the presence of thin structures, different textures and diverse shapes. Furthermore, humans can deform their facial geometry in a remarkable way, performing a wide range of expressions and emotions. Consequently, building a robust 3D face reconstruction pipeline that covers all potential states of a human face is a challenging endeavor. Several 3D face reconstruction benchmarks have been previously proposed to rank reconstruction methods in terms of quality and robustness.

In table 1 we present a comparison of popular single-image face reconstruction benchmarks. However, we find that most existing benchmarks rarely evaluate extreme facial expressions, an important aspect of human face geometry. This can be seen in fig. 3 where we retrieve the 5 most expressive images from the recent FaceScape benchmark Zhu et al. (2023) and the established NoW benchmark Sanyal et al. (2019). We do this by running EMOCA Daněček et al. (2022) on each image of the dataset, collecting the expression codes, and then performing furthest point sampling in EMOCA's expression space, starting from the expression with highest norm. We find that FaceScape only contains 20 different but relatively articulated expressions while the NoW benchmark is dominated by mostly neutral and smiling expressions. We therefore propose a new benchmark for 3D face reconstruction that is sourced from images of the recently published multi-view video dataset NeRSemble Kirschstein et al. (2023). For 21 diverse identities, we select 20 distinct expressions via furthest point sampling in expression space, for a total of 420 images. The corresponding ground truth 3D geometries are obtained by running COLMAP Schönberger & Frahm (2016) on the 16 full resolution 3208x2200 images. Additionally, we compute one pointcloud for a neutral frame of each person, yielding 441 ground truth 3D geometries in total.

### A.1.1  Task Description

Our benchmark consists of two 3D face reconstruction tasks, given a single image as an input: *posed* and *neutral* 3D face reconstruction. It is the first benchmark that evaluates both settings at the same time. The following briefly defines the differences of both tasks.

**Posed Reconstruction:**  The posed reconstruction task aims to measure the fidelity of a 3D reconstruction. Given an image of a face under arbitrary facial expression, the underlying geometry shall be recovered. This requires images with paired ground truth geometries which are available in NeRSemble trough COLMAP.

**Neutral Reconstruction**  The neutral reconstruction task on the other hand is specific to the face domain and measures how well a reconstruction method can disentangle the effects of shape and expression on a human 3D face. Specifically, the task is to reconstruct the geometry of a person's face under neutral expression given an image of the person under any arbitrary expression. Hence, the reconstruction method needs to understand the current facial expression, how it deforms the geometry and how the face would look like under neutral expression. On the other hand, this task does not *explicitly* measure whether a method can reconstruct expressions well.

**Comparison to Existing Benchmarks**  The two established benchmarks from Feng et al. (2018) and Sanyal et al. (2019) capture images and a 3D scan separately, therefore the observed expression does not match the ground truth geometry. As a consequence, these benchmarks can only measure *neutral* reconstruction performance. In contrast, two other recent benchmarks ((Zhu et al., 2023; Chai et al., 2022)) merely evaluate posed reconstructions. Our benchmark is the first to evaluate both tasks at the same time.

|  | Neutral | | | Posed | | |
|---|---|---|---|---|---|---|
|  | L1↓ | NC↑ | R²·⁵↑ | L1↓ | NC↑ | R²·⁵↑ |
| Ours + PosMap | 1.70 | 88.1 | 90.7 | 2.21 | 87.3 | 0.857 |
| Ours, + $\mathcal{N}_{\text{neutral}}$ | 1.71 | 88.2 | 90.5 | 1.73 | 87.3 | 90.4 |
| Ours | **1.66** | **88.3** | **91.2** | **1.66** | **88.4** | **91.6** |

Table 6: **Ablations of Different Prior Modalities:** We ablate the effect of extending our optimization energy $E$ (see eq. (7)) with additional priors.

### A.1.2 EVALUATION PROTOCOL

To measure the performance of a reconstructed posed or neutral 3D face, we follow established practice and first rigidly align the prediction to the ground truth point cloud via landmark correspondences and ICP. Furthermore, we use segmentation masks Zheng et al. (2022) to remove non-facial areas (hair, neck, ears, and mouth interior) from the ground truth. We then compute three metrics: (i) uni-directional L1 Chamfer distance from GT points to the nearest mesh surface, (ii) cosine similarity (NC) of predicted mesh normals and GT point cloud normals, and (iii) Recall thresholded at 2.5mm ($R^{2.5}$) which is the percentage of GT points whose nearest mesh surface is 2.5mm or closer.

### A.2 ABLATIONS

In fig. 7, we present qualitative results corresponding to our quantitative ablation study in table 4. Note that we focus on *posed* reconstructions, since the *neutral* reconstruction quality is heavily aided by the MICA prediction.

### A.2.1 ADDITIONAL PRIORS

Next to the prediction of surface normals and UV-coordinates, as presented in the main paper, it is possible to predict different modalities. In particular, we also studied the prediction of 3D position maps in the canonical coordinate frame of the face, and the prediction of surface normals in *neutral* space. We present a quantitative comparison in table 6, and a detailed description in the following two paragraphs:

**Canonical Position Map Prediction** Predicting depth or position maps similar to DUSt3R Wang et al. (2024) is another natural choice of a generic geometric cue, next to surface normals. Due to possibility to define an unambiguous canonical coordinate frame for faces, we find that predicting per pixel 3D position in that canonical reference frame is more suitable than depth prediction, which heavily depends on the camera position. We thus define the network

$$\mathcal{P} : \mathbb{R}^{512 \times 512 \times 3} \to \mathbb{R}^{512 \times 512 \times 3} \tag{9}$$

similar to $\mathcal{N}$ in eq. (1), and train it similar according to eq. (2). Data pre-processing is also conducted in a similar manner, by simply rendering vertex positions instead of normals. Although the prediction position maps look reasonable, integrating a position map rendering loss

$$\mathcal{L}_p = \|\mathcal{P}(I) - \texttt{render}_p(V)\| \tag{10}$$

into our optimization energy eq. (7) turns out to deteriorate reconstruction quality, as shown in table 6.

**Neutral Surface Normals** Similar to $\mathcal{N}$ in eq. (1), it is possible to define a pixel-aligned surface normal estimation task aimed to help disentanglement of identity and expression. To this end, we define $\mathcal{N}_{\text{neutral}}$ which predicts per-pixel normals in *neutral* space, as opposed to $\mathcal{N}$ which predicts posed-space surface normals. However, ground truth for per-pixel neutral normals is unknown for 3D scans. Therefore, we resort to our non-rigid registration results. To obtain neutral normals, we render per-pixel bary-centric coordinates of any

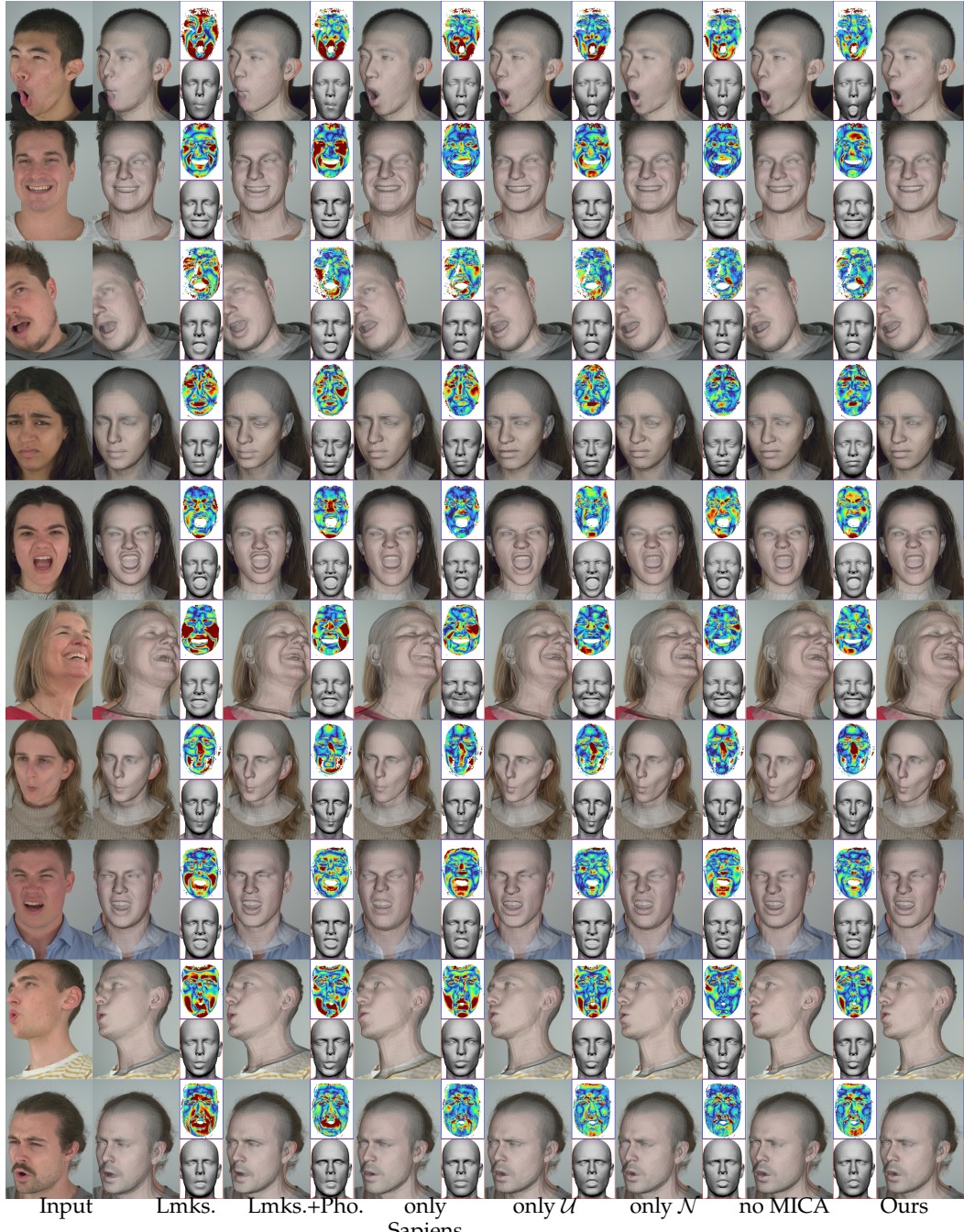

Input  Lmks. Lmks.+Pho. only  only $\mathcal{U}$ only $\mathcal{N}$ no MICA Ours
           Sapiens

Figure 7: **Ablation Study (Posed):** We present a qualitative comparison to several ablation experiments.

registered posed mesh, which allows us to index the registered neutral mesh of the same person in order to determine the neutral surface normal.

Once trained, we extend our optimization energy eq. (7) by

$$\mathcal{L}_n^{\text{neutral}} = \|\mathcal{N}_{\text{neutral}}(I) - \text{render}_n(V_{\text{neutral}})\|, \tag{11}$$

where neutral vertices $V_{\text{neutral}}$ are obtained with FLAME parameters for which all attributes except for the shape parameters have been set to zero. Doing so, however, slightly impairs reconstruction quality, as shown in table 6. While the neutral scores are impaired less,

|  | Neutral | | | Posed | | |
|---|---|---|---|---|---|---|
|  | L1↓ | NC↑ | $R^{2.5}$↑ | L1↓ | NC↑ | $R^{2.5}$↑ |
| Ours, Single Image | 1.51 | 88.2 | 92.7 | 1.53 | 87.4 | 93.3 |
| MICA, frame average | 1.46 | 88.3 | 94.6 | - | - | - |
| Ours, Monocular Video | **1.38** | **88.4** | **96.4** | **1.45** | **88.2** | **94.8** |

Table 7: **Ablation on Observation Density:** Increasing the observation density by extending our optimization over a monocular video sequence improves reconstruction results. Especially, neutral reconstruction performance benefits from multiple observation of the same person under changing facial expressions and head poses.

the prediction quality of $\mathcal{N}_{\text{neutral}}$ is not good enough. We speculate that predicting neutral normals is more prone to overfitting, since the task becomes more ill-posed and our training dataset consists of only a relatively small number of identities. Furthermore, small errors introduced in our registration procedure lead to a more noisy training signal. Note that for training $\mathcal{N}$ we can leverage ground truth 3D scans, instead of registrations thereof.

### A.2.2 MONOCULAR VIDEO TRACKING

In another ablation experiment we analyze the effect of the observation density on the reconstruction quality. To this end we compare single image reconstruction quality against monocular video reconstruction results. Due to the rather static head poses of the NeRSemble Kirschstein et al. (2023) video recordings, we select the five videos with the most significant head movement which were included in our single image reconstruction benchmark. We then select the a frontal camera and compare how reconstructions change, when using the whole video sequence as input, compared to just using individual frames. Quantitative results are presented in table 7. We notice that especially neutral reconstruction quality benefits, from including multiple observations of the same person under changing expressions and head poses. For a more complete comparison, we include the evaluation of frame-averaged MICA Zielonka et al. (2022) predictions, which serve as initialization to our tracking. The results indicated that our optimization significantly improves upon MICA due to a higher observation density. This experiments shows that our neutral reconstruction performance significantly improves in a video tracking scenario with sufficient (and potential not too extreme) head rotation.

### A.3 FAILURE CASES

While our method generally performs very robust, even with respect to extreme head rotations, lighting conditions and occlusions, we find that certain, extreme facial expression cannot be properly represented. We show such examples from our proposed benchmark in fig. 9. The fitting inaccuracies are partially caused by our prior networks, and partially caused by FLAME's inability to represent complex lip movements.

In general, our normal estimator $\mathcal{N}$ seems to generalize the best to such out of distribution expressions. In contrast, our UV-coordinate prediction network $\mathcal{U}$ is already limited by its training data, which has been obtained using our registration procedure combining FLAME fitting with non-rigid-registration. Therefore, if such expressions are in our training dataset, their registrations will likely not be perfect, which reflects in the prediction quality. Similarly, our reconstruction procedure is likely to fail such expressions due to its dependence on the FLAME model. In the future, we hope to see similar approaches, which utilize more powerful 3DMMs, such as MonoNPHM Giebenhain et al. (2024).

Furthermore, we illustrate the error distribution of the 1701 examples in the NoW test set in fig. 8. As can be seen, the mean error, measured in mm, per example follows a well behaved distribution. There are merely 21 examples with

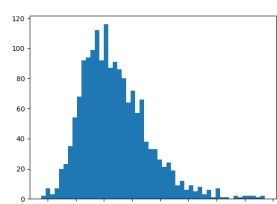

Figure 8: Error Distribution

an error higher than 1.8mm and 8 have an error higher than 2.0mm.

Finally, we encourage the reader to watch our supplementary video and supplementary tracking comparison to MetricalTracker Zielonka et al. (2022) for examples that showcase the robustness of our approach.

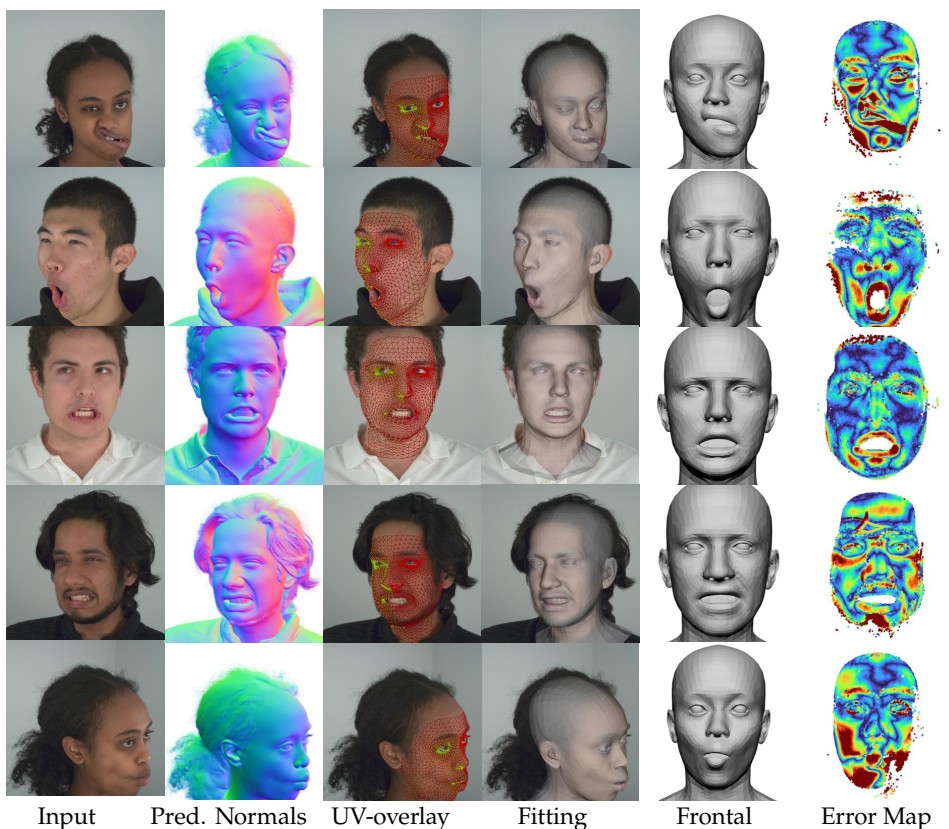

| Input | Pred. Normals | UV-overlay | Fitting | Frontal | Error Map |

Figure 9: **Failure Cases:** Extreme expressions can pose an issue to our method, which is mainly caused by the low representation capacity of the FLAME model. Thus, inference-time optimization is impeded. The same holds for obtaining high-fidelity and consistent registrations to get g.t. UV-coordinates for training.

## A.4    Surface Normal Estimation

In this section we provide additional qualitative surface normal estimation results.

### A.4.1    In-the-Wild Normal Estimation

One central quality of surface normal estimators, which ultimately makes them valuable to our community, is their generalization to arbitrary in-the-wild images of human heads. Thus, fig. 11 provides additional estimation results on the FFHQ dataset (Karras et al., 2019). Here, we qualitatively compare against results of two recent industry foundational models, Sapiens (Khirodkar et al., 2024) and DAViD (Saleh et al., 2025), as well as, Diff-E2E (Martin Garcia et al., 2025), another recent surface normal estimator distilled from StableDiffusion (Rombach et al., 2022). While Sapiens, tends to produce blurry results, DIff-E2E over-emphasizes geometric details, resulting in unnatural sharp edges. Finally, DAViD, a concurrent work to ours, produces the most competitive results on such images. However, similar to our findings in the main paper, the synthetic training data of DAViD is clearly noticeable in its predictions. This can especially be seen in the estimated geometry in the eye region, and the flatness of predicted wrinkles and creases of the skin. Pixel3DMM

tends to produces the most visually pleasing results, while all methods produce fairly robust estimates.

### A.4.2 Additional Visualizations

Furthermore, we provide additional visualizations in fig. 12, which corresponds to the quantitative evaluation in table 5. Here, we also show error maps, visualized using a `turbo`-coloring scheme. Please note that the camera registration in the H3DS dataset Ramon et al. (2021) is slightly misaligned, which results in significantly higher errors, compared to the NeRSemble Kirschstein et al. (2023) and MultiFace Wuu et al. (2022) evaluations. The error maps confirm our error analysis of the previous subsection. In general, all methods perform similar on in-the-wild and studio images, confirming their generalization abilities.

### A.5 Additional Baselines

Finally, we present qualitative comparisons to VHAP (Qian et al., 2024) and Neural Head Avatars (NHA) (Grassal et al., 2022) in fig. 13. While both methods were originally designed for monocular video tracking, they can still be executed on a single image. However, the increased sparsity of a single observations results in poor 3d reconstructions. In particular, both approaches consist of a two stage reconstruction paradigm: In the first stage reconstruction is performed in the FLAME latent space, which heavily regularized the optimization problem. In the second stage, both methods optimize for per-vertex offsets, which increases the representational capacity. Especially, the second stage tends to overfit to the single-view observation and degrade 3d accuracy.

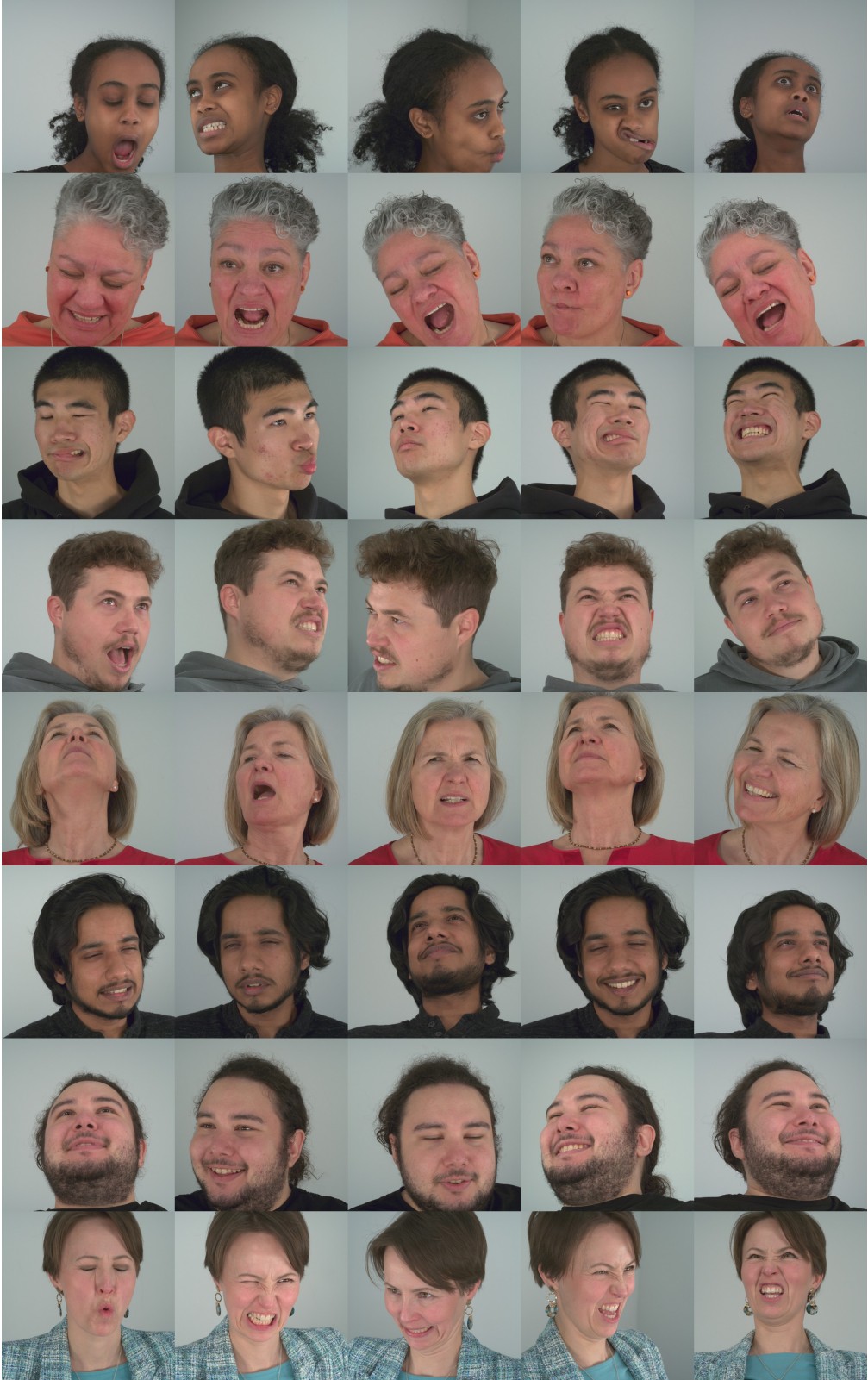

Figure 10: **Benchmark Overview:**

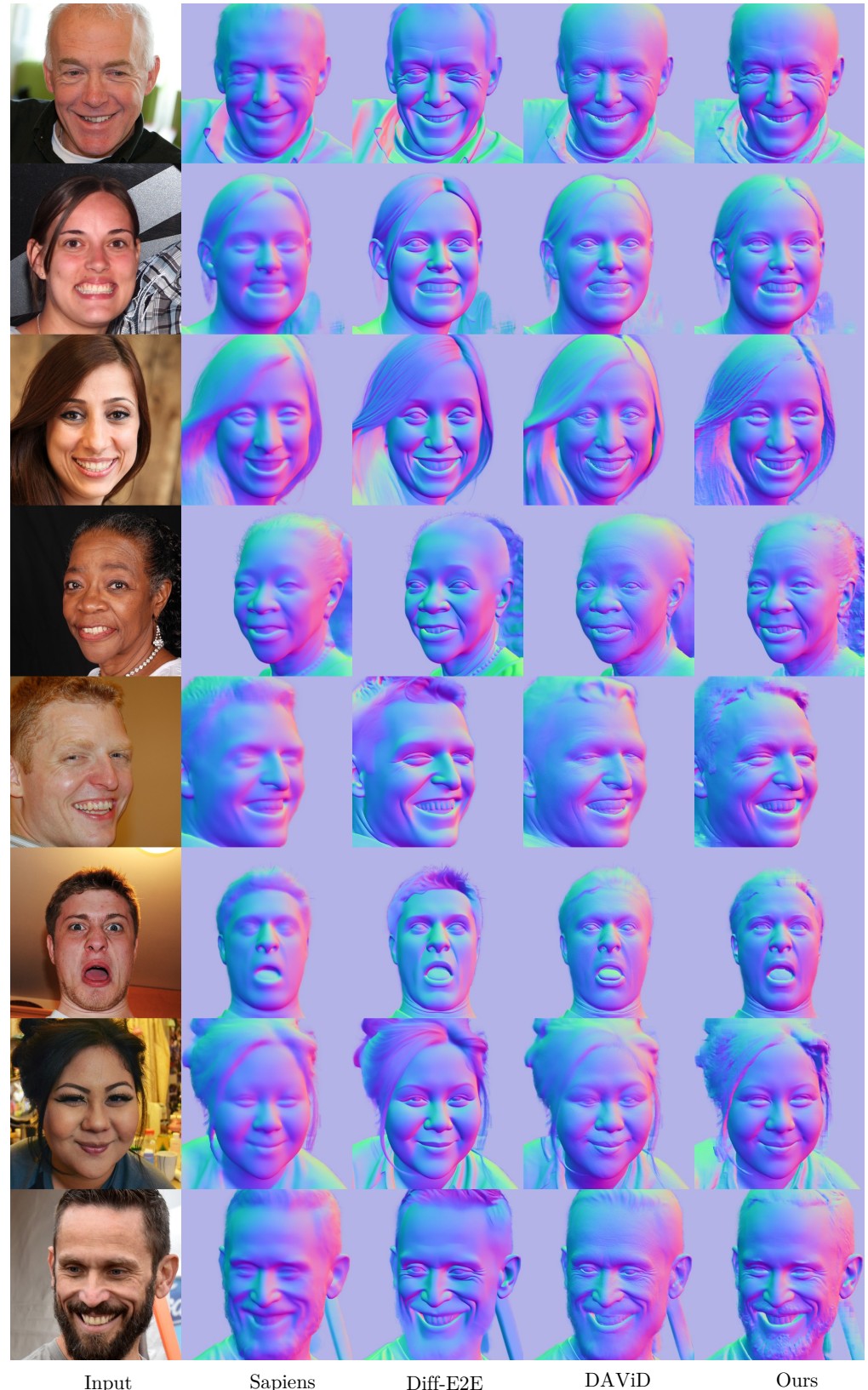

Input          Sapiens          Diff-E2E          DAViD          Ours

Figure 11: **Surface Normal Estimates on FFHQ.**

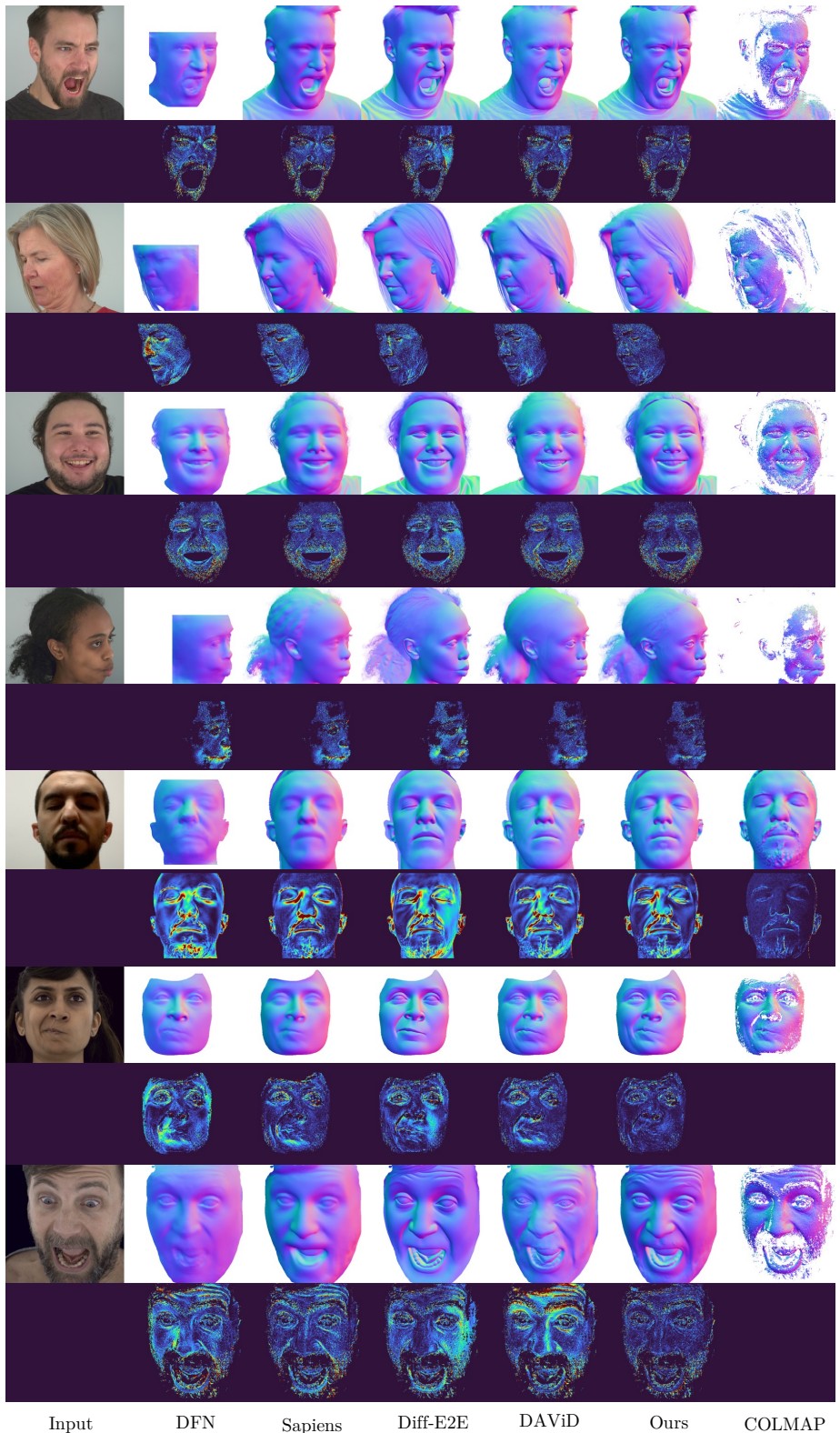

Input     DFN     Sapiens     Diff-E2E     DAViD     Ours     COLMAP

Figure 12: **Surface Normal Estimation:** The first four rows show results from NeRSemble, fllowed by one example from H3DS and two examples from MultiFace.

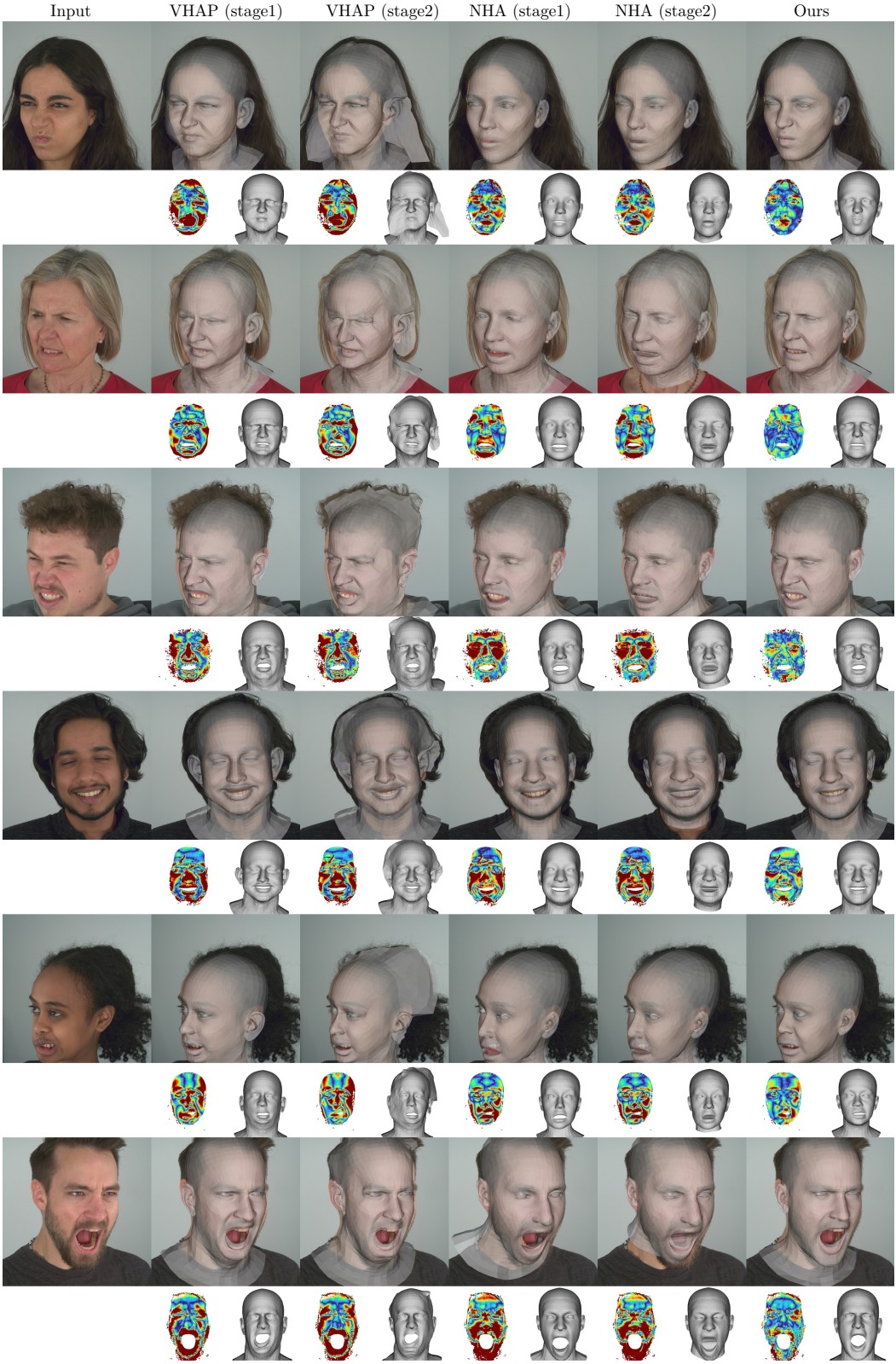

Figure 13: **Additional Baselines:** Posed reconstruction.

