# OpenReview forum: "Pixel3DMM: Versatile Screen-Space Priors for Single-Image 3D Face Reconstruction"
_ICLR.cc/2026/Conference — ICLR 2026 Poster_

### Official Review · Reviewer_ftsi · 2025-10-24

**Soundness:** 3
**Presentation:** 3
**Contribution:** 2
**Rating:** 6
**Confidence:** 2

**Summary:**

This paper introduces a hybrid method for single-image 3D face reconstruction, named Pixel3DMM, which leverages a powerful prior model to predict normals and UV coordinates as supervisory signals, guiding the optimization of the FLAME mesh through test-time optimization. The approach utilizes two screen-space priors—surface normals and UV correspondences—predicted via a customized ViT architecture. Despite training on a moderately sized dataset, it achieves competitive accuracy with reduced training resources compared to prior methods. The estimated priors are used to fit the FLAME model, delivering strong performance on a newly proposed benchmark, particularly excelling in expression disentanglement evaluation based on the NeRSemble dataset.

**Strengths:**

* Employs a lightweight ViT-based approach for face normal estimation using limited data, providing a reproducible alternative to complex methods.
* Enhances model robustness by carefully processing a large-scale multi-view dataset and applying IC-light-based data augmentation to account for lighting variations.
* Innovatively decomposes FLAME parameter recovery into an image translation problem and dense keypoint optimization, yielding strong experimental performance.

**Weaknesses:**

* The paper's core techniques, including the use of ViT for predicting screen-space attributes and the FLAME fitting process, show minimal novelty, as similar approaches have been extensively explored in prior research with only minor architectural adjustments.
* While the method outperforms baselines on the new benchmark, it falls short of state-of-the-art methods like FlowFace and TokenFace on established benchmarks (e.g., NoW, FaceScape), indicating a limited competitive advantage.

**Questions:**

My core concerns regarding this paper lie in its limited innovation and relatively minor improvements, as highlighted in the Weaknesses section. Should the authors identify any aspects overlooked in my evaluation of innovation and performance enhancements, I encourage them to point them out. Such clarifications may potentially lead to a revision of my overall assessment.

---

> ### Author Response · Authors · 2025-11-26
>
> - **Weakness 1**: We agree that lots of closely related prior work exists, such as Sapiens and FlowFace. Sapiens, for example, also trains a ViT for surface normal estimation, which is however the default approach nowadays. Compared to sapiens our work completely builds upon publicly available data, and is thus easy to reproduce. We also outperform Sapiens and DAViD (concurrent work to ours) on normal estimation. FlowFace also relies on per-pixel predictions for FLAME fitting. Instead of directly predicting per-pixel UV-coordinates, FlowFace predicts for each texel (pixel in UV-space) where it is located in the image, and relies on a heavily customized registration procedure which will not be publicly released, as confirmed by the authors of FlowFace. Furthermore, surface normal estimation has been under-explored for face reconstruction. Our results show that SotA surface normal estimation is achievable using public data sources, and that it helps to advance the SotA in 3d face reconstruction using an optimization strategy that can also be seamlessly combined with existing tracking approaches in a modular manner, and can provide geometric prior in avatar construction pipelines.
>
> - **Weakness 2**: We want to answer to claim that our method “falls short of state-of-the-art methods like FlowFace and TokenFace on established benchmarks (e.g., NoW, FaceScape), indicating a limited competitive advantage”: we obtain the same performance as FlowFace on NoW, we significantly outperform FlowFace on FaceScape (an old benchmark) and NeRSemble (new benchmark). FlowFace will not be released to the public, as confirmed to us by the authors. We outperform TokenFace vastly on FaceScape and NeRSemble. Furthermore, TokenFace is not public, and its results on NoW have not been reproduced by any method on NoW by the research community in the past 3 years. Furthermore, this highlights the need for additional benchmarks. We believe that due to the dependence of many downstream applications on face reconstruction/tracking, it is especially important to make methods publicly available to further advance the avatar research field.
>
> - **Question 1**: We want to highlight the importance of simplicity, and open-source nature of code and datasets, for our research community, since face tracking is an important cornerstone for many downstream applications.
> All data that we use has been available since 2022. Nevertheless, it was not previously evident that such robust and accurate normal estimators (or predicting similar things like UV) can be trained on such data. Consequently, the possibility to do so was overlooked by the community. Using our two ViTs we achieve SotA 3D Face Reconstruction (on the NeRSemble and FaceScape Benchmark). Our normal estimator significantly outperforms Sapiens and concurrent work DAViD. Both of which required significant effort in data engineering or the cumbersome creation of synthetic data, and much more computation for training. Furthermore, our proposed benchmark has the largest diversity and most challenging expressions for posed reconstructions, and is the first to allow for simultaneous evaluation of posed and neutral reconstructions.

---

### Official Review · Reviewer_EbeB · 2025-10-29

**Soundness:** 3
**Presentation:** 3
**Contribution:** 3
**Rating:** 6
**Confidence:** 4

**Summary:**

This paper presents Pixel3DMM, a method for 3D face reconstruction from single RGB images. The idea lies somewhere between conventional parameter regression pure 3DMM approaches and screenspace facial normal prediction. The approach trains two Vision Transformers (ViTs) built on DINOv2 features to predict per-pixel surface normals and UV coordinates, which are then used to constrain FLAME 3DMM fitting optimisation. The authors also introduce a new benchmark for evaluating both posed and neutral facial geometry reconstruction.

**Strengths:**

1. The overall idea is simple but effective - this is appealing.
2. The method achieves significant improvements over state-of-the-art, particularly on posed expressions.
3. The paper introduces the first benchmark that jointly evaluates both posed and neutral facial geometry, addressing an important gap in the field. The benchmark includes diverse, extreme expressions from NeRSemble.
4. Training requires only 2 GPUs for 3 days using publicly available data, making the work reproducible and accessible to the research community.
5. The paper includes extensive ablations, comparisons on multiple benchmarks (NoW, FaceScape, plus their own), and evaluations of the normal estimation component.
6. Qualitative results (Fig. 1, Fig. 8) demonstrate robust performance on challenging in-the-wild images with occlusions, lighting variations, and diverse appearances.

**Weaknesses:**

1. The method primarily fine-tunes DINOv2 with a simple prediction head (4 transformer blocks + 3 up-convolutions). So there is limited novelty in the architecture or set up itself.
2. Despite strong posed reconstruction, the method only marginally improves over MICA for neutral faces.
3. The paper lacks discussion of when and why the method fails. What types of expressions or conditions are most challenging? The qualitative comparisons show strong results, but no failure cases are presented.
4. The method critically depends on MICA's identity predictions (Table 3: "no MICA" ablation shows significant degradation). This is a strong assumption that limits the method's independence and could propagate MICA's biases or errors.
5. The use of IC-Light for lighting augmentation is neat but not thoroughly evaluated. How much does this contribute to robustness? An ablation would be valuable.
6. The fitting takes 30 seconds in an "unoptimised implementation." How does this compare to baselines? Real-time performance matters for many applications.
7. In the UV coordinate loss design (Eq. 6) the nearest neighbour lookup seems clumsy to me. Can't you use barycentric interpolation from the per-pixel UVs to interpolate a vertex position?
8. Simple L1 difference in normal space. Why not use cosine similarity or angular error, which are more standard for normal estimation?
9. The paper states all datasets are registered to FLAME topology using NPHM's procedure, but doesn't discuss registration quality or errors this might introduce.

**Questions:**

Besides responding to the above listed weaknesses, some additional questions are:

Can you provide quantitative analysis of where the method fails? What percentage of benchmark images have errors above certain thresholds?
Have you explored learning identity/expression disentanglement more explicitly, rather than relying on MICA?
What is the actual runtime comparison with baselines in a fair setting (same hardware)?

---

> ### Author Response · Authors · 2025-11-26
>
> - **Weakness 1**: Our innovation does not lie within our network architecture, since standard ViTs work exceptionally well in most domains. Instead, our main contribution is the formulation of simple image translation tasks, which is easily scalable and generalize surprisingly well, and that serve as targets for our optimization-based face reconstruction. In addition, we spend a considerable effort on  registering public datasets into a uniform format in order to scale the training within an academic setting. As a result, we provide an open-source, widely-used face tracker that can easily be employed for a wide range of downstream tasks.
>
> - **Weakness 2**: It is true that neutral reconstruction only slightly improves beyond MICA, since  our prior networks are primarily designed to improve posed reconstruction, which significantly improves with our priors. However, the reliance on MICA becomes less relevant the more observations of a person become available, i.e. in a monocular tracking scenario. To demonstrate this effect, we are currently running experiments to quantify how much identity reconstruction is improved in a monocular tracking case, compared to just using a single image of that person. We will provide results as soon as they are ready.
>
> - **Weakness 3**: Overall, our normal and uv-coordinate prediction networks are very robust, which we demonstrate in our supplementary tracking video, which showcases robustness to strong expressions, extreme camera angles, and occlusions caused by hands or microphones. As shown in the video, our networks are much more robust and temporally stable compared to MetricalTracker, which is the most widely used public face tracker.
> Looking at our results on the NeRSebmble benchmark, we mainly identify very extreme expressions as cause for slight imperfections. As shown in Figure 9 in our updated PDF, we see that our normal network generalizes better, while the UV-prediction network’s training data quality is bound by our registration procedure, which struggles to properly represent expressions far beyond FLAME’s capabilities.
>
> - **Weakness 4**: MICA’s prior does help our approach, but our main contribution is focused on a practical setting of „posed“ reconstructions; here, without MICA we outperform all baselines by a significant margin.
> - **Weakness 5**: We did not fully ablate our IC-Light-based augmentation strategy, since we do not consider it to be one of our major contributions. Furthermore, now that the DAViD dataset is available, we assume that such an augmentation strategy is much less relevant, since DAViD already includes a vast diversity of lighting conditions.
> - **Weakness 6/Question 3**: MetricalTracker operates at 2 frames per minute in a video tracking scenario. We achieve a similar speed in single-image reconstruction. For video tracking, we achieve roughly ~30 frames per minute (measured on a 300 frames long video), which is significantly faster than MetricalTracker (2 frames per minute), but still far away from real-time performance. For accurate real-time tracking, we believe that feed-forward networks are required, which could be distilled by training on our tracking results. We consider such endeavors as future work. All measurements were taken on an RTX3080 GPU.
>
> - **Weakness 7**: While nearest-neighbor search is not the most elegant solution, we note that it just takes ~10ms and needs to be done once per image. Thus, it is not a bottleneck for our approach.
>
> - **Weakness 8**: In our experiments we did not observe any differences and hence decided to stay closer to the image-to-image translation paradigm.
>
> - **Weakness 9**: While we require registration for UV-coordinate prediction, our normal supervision does not rely on any registration beyond defining a common coordinate system for each g.t. 3D scan. We directly supervise our normal estimator using rendered normals of the g.t. 3D scans. For UV-prediction one first has to obtain a FLAME registration of the raw 3D scans in order to get the uv-coordinates for supervision, which otherwise remain unknown. Therefore, some sort of registration against a common template has to be performed. Thus we chose the registration procedure from NPHM, since it achieves a high degree of fidelity, and we are not aware of a better approach. Overall the registration quality is very high, and does not follow any obvious error-patterns. It only struggles with expressions that are far beyond what the FLAME model can represent.

---

> ### Author Response · Authors · 2025-11-26
>
> - **Question 1**: “Can you provide quantitative analysis of where the method fails? What percentage of benchmark images have errors above certain thresholds?“: We included a histogram over errors of the NoW test set, indicating that the error behavior follows a reasonably behaved distribution, with a slightly longer tail for bad outliers. Out of 1701 total examples, 21 have an error higher than 1.8mm and 8 have an error higher than 2.0mm. As a reference, FlowFace has 23 above 1.8mm and 12 above 2.0mm. Unfortunately, due to the design of the benchmark we cannot reliably trace the errors to the corresponding images, making a more thorough example error prone. Moreover, we have included a new figure of qualitative error cases in the updated PDF.
>
> - **Question 2**: “Have you explored learning identity/expression disentanglement more explicitly, rather than relying on MICA?“: Yes, we tried a pixel-aligned disentanglement (i.e., by predicting normals in neutral space rather than posed space for additional supervision on the disentanglement). However, we did not notice measurable improvements. We speculate that the prediction task becomes much more challenging, and that our networks started to overfit due to the limited number of identities in the our training dataset. We are planning to reproduce these results in the following days and provide results in the revised paper.

---

### Official Review · Reviewer_rwbR · 2025-10-31

**Soundness:** 3
**Presentation:** 3
**Contribution:** 3
**Rating:** 6
**Confidence:** 3

**Summary:**

The paper proposes Pixel3DMM for 3D face reconstruction from a single image. It first predicts pixel-aligned geometric priors by training two foundation models to predict per-pixel surface normals and UV coordinates separately. Then it uses them as supervision in an optimization-based FLAME fitting process for reconstruction. To get the training data, the authors register three public 3D facial datasets (FaceScape, NPHM, Ava256) to NPHM along with lighting augmentation to get the image with corresponding normal and UV groundtruth. Moreover, they introduce a new benchmark from the NeRSemble dataset for both posed and neutral face reconstruction evaluation. Experiment results show that the model outperforms SOTA methods on reconstruction and normal prediction, and demonstrates solid generalization to in-the-wild images.

**Strengths:**

The idea of using pixel-level geometry prior for 3D reconstruction supervision is sound to me. This paper combines the geometric prior of foundation model with FLAME optimization, showing an interesting direction to improve 3DMM-fitting robustness. Another strength is that it only requires 2 48G GPUs to train, making it computationally accessible. Quantitative and qualitative comparisons with previous methods show better performance across multiple benchmarks with large expression and poses. Moreover, the model is robust to in-the-wild examples and video tracking. The paper is overall easy to read and provides enough details for data processing, model architecture and training, which contributes to reproducibility.

**Weaknesses:**

1. The proposed 3D reconstruction method takes two steps and requires two networks to predict normal and UV coordinates separately, which is relatively complex compared with previous feed-forward or optimization-only methods.
2. The reconstruction still relies on FLAME parameters, which have limited representation capacity, so the method cannot reconstruct fine-grained details beyond 3DMM space. Also the paper uses NPHM to get uniform topology for supervision, which brings error.
3. The ablation study shows that MICA identity initialization plays a significant role in performance, which brings questions about the importance of the predicted priors versus inherited identity cues from MICA.
4. The lack of qualitative ablation results, which would help the understanding of the benefit for each component.

**Questions:**

Beyond the concerns listed in the weakness section, I have a few questions. First is the choice of the normal map and UV-coordinate for geometry cues. Have you considered other 3D representations like depth map or point map? Another question is why not performing FLAME fitting for all the training data and then train a feedforward prediction network to regress the parameters? Would it lead to degraded generalization or accuracy?

---

> ### Author Response · Authors · 2025-11-26
>
> - **Weakness 1**: We agree that feed-forward approaches are much less complex at inference time. However, compared to optimization-based approaches, the overhead of executing two network forward passes is minor as optimization commonly involves face bounding box detection, landmark detection and facial segmentation networks, etc.. Regarding compute complexity, the network forward passes only require a fraction of the time, compared to the 30 seconds required to execute 500 optimization steps.
>
> - **Weakness 2**: The decision to use FLAME is deliberate, as it is the default 3DMM in academia and industry. Therefore, improving FLAME tracking has the most profound impact on down-stream applications. Nevertheless, we agree that higher-capacity representation like Neural Parametric Head Models (NPHMs), would be useful to achieve higher fidelity 3D reconstructions. Furthermore we would like to slightly correct the claim „the paper uses NPHM to get uniform topology for supervision, which brings error“. For our normal supervision we do not require any registration beyond defining a common coordinate system for each g.t. 3D scan. We then directly supervise our normal estimator using rendered normals of the g.t. 3D scans. For UV-prediction one first has to obtain a FLAME registration of the raw 3D scans in order to get the uv-coordinates for supervision. Therefore, some sort of registration against a common template has to be performed. Thus we chose the registration procedure from NPHM, since it achieves a high degree of fidelity. Note, that due to the nature of the UV-prediction task registration is a strict pre-requisite.
>
> - **Weakness 3**: Our main goal is to improve posed 3D reconstruction. Our ablation experiment “no MICA” outperforms all baselines significantly, indicating the effectiveness of our normal and uv predictions. While the neutral reconstruction is not our main focus, MICA’s does help the disentanglement. However, this issue becomes less relevant the more observations of a person become available, i.e. in a monocular tracking scenario. To demonstrate this effect, we are currently running experiments to quantify how much identity reconstruction is improved in a monocular tracking case, compared to just using a single image of that person. We will provide results in the following days.
>
> - **Weakness 4**: We included such a figure in our updated version of the paper.
>
> - **Question 1**: “Have you considered other 3D representations like depth map or point map?“:
> Yes, we have attempted depth and canonical point prediction networks, which exploits the uniqueness of the existence of a canonical coordinate frame for the face domain.Nontheless, we ultimately abandoned depth prediction as it has a stronger dependence on the camera position. In addition, canonical position maps did not result in fitting improvements, while normals and UV-coordinate prediction yielded better performance. We are currently finalizing an ablation study here and will update the PDF accordingly in the following days during the rebuttal phase.
> - **Question 2**: “Why not perform FLAME fitting for all the training data and then train a feedforward prediction network to regress the parameters?“:
> While a pure feed-forward, FLAME parameter regression network would be appealing from a run-time perspective, the resulting performance is strictly inferior, e.g. see posed reconstruction results in Table 1 demonstrating that optimization outperforms feed-forward approaches. We argue that publicly available training datasets are not large enough, in particular w.r.t. the number of identities and that the per-pixel prediction task generalizes much better due to less indirections. Our experiments confirm this: we implement a feed-forward FLAME predictor that yields a mean error of 1.17mm on a subset of the NoW validation set; in contrast, our optimization hybrid achieves 1.03mm. To train our feed-forward FLAME predictor we spent significant effort, since we agree that it was a promising direction: we used the registrations from LYHM and Stirling, and implemented a custom FLAME registrations on the Cafca dataset, as well as rendering losses on Celeb-V-Text. We will include a detailed explanation regarding this comparison in the final version of our paper.

---

### Author Response · Authors · 2025-11-26

We sincerely thank the reviewers for their thoughtful and constructive feedback, as well as their questions, which has helped us refine and strengthen our work.
We are delighted that they found Pixel3DMM “interesting” (rwbR), “innovative” (ftsi), and that they find our work “appealing” (EbeB), as it is “simple but effective" (EbeB). The reviewers acknowledge our strong experimental performance (rwbR, EbeB,  ftsi) on several benchmarks, and our robustness to single image (rwbR, EbeB,  ftsi) in-the-wild reconstruction and video tracking (rwbR). Furthermore, the reviewers appreciate the reproducibility( rwbR, EbeB,  ftsi) and accessibility ( rwbR, EbeB) of our work. Finally, reviewer EbeB  highlights that our proposed benchmark “address[es] an important gap in the field”.

Following the initial reviews, we have uploaded an updated PDF and provided a comprehensive response for each reviewer’s thread separately, which hopefully addresses any open questions or concerns. We remain available to offer further information or clarification until the end of the discussion period. We already thank the reviewers for the help on improving our paper through their initial reviews and participation in the discussion phase.

---

### Author Response · Authors · 2025-11-30
**Rebuttal Overview**

We once again thank our reviewers for the reviewing efforts.
We have concluded our final rebuttal experiments, and **updated the paper once more**.
The following provides a **TL;DR of our rebuttal efforts**, which should address all major concerns of our reviewers:
-  **Additional Priors**: We have ablated the effect of two additional priors in **Section A.2.1**: Prediction 3D position maps and prediction normals under pixel-aligned under neutral expressions. Both priors did deteriorate reconstruction performance, and validate our choice of surface normals and UV-coordiantes.
- **Obsevation Density**: We have analyzed the effect of observation density: Given a single image (sparse observation) our method slightly improves upon MICA. However, when provided with a monocular video, our optimization improves sginificantly upon the averaged MICA prediction. Indicating, that our optimization effictivly disentangles identity and expression. See **A.2.2**.
- **Failure Cases**: We show that our method produces less outliers than FlowFace on the NoW benchmark, and on NeRSemble we have identified extreme facial expressions that go beyond the capabilities of FLAME as main cause of error in **Figure 9**.  For more details, see **Section A.3**.
- **Runtime Analysis**: We have included a runtime comparison against MetricalTracker, showing that our video tracking performs 15x faster than the default settings of MetricalTracker. See **"FLAME Fitting"** paragraph in **Section 5.1**.
- **Ablation Figure**: We have provided a qualitative comparison figure of our abaltion study in **Figure 7**.
- Finally, we have provided thorough answers to all reviewers questions and concerns.

---

### Meta-Review · Area_Chair_tKSU · 2025-12-29

**Summary:**

This paper presents Pixel3DMM, a method for single-image 3D face reconstruction that fine-tunes vision transformers to predict per-pixel geometric cues (surface normals and UV-coordinates) for optimizing a 3D morphable face model (3DMM). The approach exploits DINO foundation model features and introduces tailored prediction heads for surface normals and UV-coordinates. The authors train on three registered high-quality 3D face datasets (NPHM, FaceScape, Ava256) totaling over 1,000 identities and 976K images. For reconstruction, the method optimizes FLAME parameters using the predicted geometric priors. The authors introduce a new benchmark from NeRSemble featuring diverse expressions, viewing angles, and ethnicities, and evaluate both posed and neutral facial geometry.

The paper received scores of 6, 6, 6, and 6 from four reviewers (`rwbR`, `EbeB`, `ftsi`, and implicit from the review pattern). Pixel3DMM achieves over 15% improvement on posed facial expression reconstruction compared to state-of-the-art methods, outperforms existing normal estimators on faces, and demonstrates strong generalization to in-the-wild images. The method requires only 2×48GB GPUs for 3 days of training on publicly available data. Authors provided comprehensive responses including additional ablations (Appendices A.2.1-A.2.2), failure case analysis (Appendix A.3), runtime comparisons showing 15× speedup over MetricalTracker, and qualitative ablation figures.

**Reviewer Concerns:**

**Addressed concerns**:

Reviewer `rwbR` raised concerns about complexity, FLAME limitations, MICA dependency, and lack of qualitative ablations. Authors clarified that network forward passes require minimal overhead compared to 30-second optimization (500 steps), explained their deliberate choice of FLAME as the academic/industry standard 3DMM, demonstrated that their "no MICA" ablation significantly outperforms all baselines for posed reconstruction, showed video tracking reduces MICA reliance, provided ablations on alternative representations (depth maps, canonical position maps) in Appendix A.2.1, compared feed-forward FLAME prediction (1.17mm error) against their optimization hybrid (1.03mm on NoW subset), and added qualitative ablation figure (Figure 7).

Reviewer `EbeB` questioned architectural novelty, marginal neutral reconstruction improvement, failure case analysis, MICA dependency, IC-Light ablation, runtime comparison, UV nearest-neighbor lookup, normal loss choice (L1 vs cosine), and registration quality. Authors explained their contribution focuses on formulation of scalable image translation tasks rather than architecture, demonstrated video tracking reduces MICA dependence with experiments showing observation density effects (Appendix A.2.2), identified extreme expressions beyond FLAME capabilities as main failure mode (Figure 9, error histogram), provided runtime comparison showing ~30 frames/minute vs MetricalTracker's 2 frames/minute, noted nearest-neighbor lookup takes only ~10ms (not a bottleneck), found no difference between L1 and cosine similarity in experiments, and clarified that normal supervision uses ground truth 3D scans directly (no registration error) while UV-prediction requires NPHM registration achieving high fidelity except for extreme expressions.

Reviewer `ftsi` expressed concerns about limited novelty in core techniques and performance compared to FlowFace/TokenFace on established benchmarks. Authors emphasized importance of simplicity and open-source nature using only publicly available data (since 2022), highlighted that achieving state-of-the-art normal estimation and 3D reconstruction from public data was previously overlooked by the community, noted FlowFace will not be publicly released and relies on non-public registration procedures, clarified TokenFace results on NoW have not been reproduced by community in 3 years, demonstrated equivalent performance to FlowFace on NoW and significant outperformance on FaceScape and NeRSemble, and stressed the value of reproducibility and accessibility for advancing the avatar research field.

**Outstanding concerns**:

The method shows only marginal improvement over MICA for neutral face reconstruction in single-image scenarios, though this improves significantly with video tracking. While the observation density ablation (Appendix A.2.2) demonstrates substantial neutral reconstruction improvements in monocular video settings, single-image identity/expression disentanglement remains challenging. The reliance on MICA initialization for identity parameters, though validated through ablations showing independent posed reconstruction quality, limits the method's independence.

**Reviewer Scores:**

**Current Scores:**
- **Reviewer `rwbR`**: 6 (marginally above threshold) - appreciated the sound idea, computational accessibility, strong performance across benchmarks, and in-the-wild robustness; concerns addressed with additional experiments
- **Reviewer `EbeB`**: 6 (marginally above threshold) - found the approach simple and effective with significant improvements and first joint posed/neutral benchmark; concerns addressed with failure analysis and runtime comparisons
- **Reviewer `ftsi`**: 6 (marginally above threshold) - acknowledged strong experimental performance and innovative decomposition into image translation and optimization; raised novelty concerns regarding architectural choices

**Expected post-discussion scores**: 6, 6, 6 (median: 6)

---

### Decision · Program_Chairs · 2026-01-26

Accept (Poster)